# Blockade of HVEM for Prostate Cancer Immunotherapy in Humanized Mice

**DOI:** 10.3390/cancers13123009

**Published:** 2021-06-16

**Authors:** Nicolas Aubert, Simon Brunel, Daniel Olive, Gilles Marodon

**Affiliations:** 1Centre d’Immunologie et Maladies Infectieuses-Paris, CIMI-PARIS, Sorbonne Université, INSERM, CNRS, 75013 Paris, France; nicolas.aubert@etu.upmc.fr (N.A.); simonbrunel@live.fr (S.B.); 2Institut Paoli-Calmettes, Aix-Marseille Université, INSERM, CNRS, CRCM, Tumor Immunity Team, IBISA Immunomonitoring Platform, 13009 Marseille, France; daniel.olive@inserm.fr

**Keywords:** immune checkpoint, HVEM, BTLA, monoclonal antibody, cancer immunotherapy, humanized mice, prostate cancer

## Abstract

**Simple Summary:**

Current immune checkpoint inhibitors have shown limitations for immunotherapy of prostate cancer. Thus, it is crucial to investigate other immune checkpoints to prevent disease progression in patients with prostate cancer. Here, we first show that the HVEM/BTLA immune checkpoint is associated with disease progression in patients. We then show that immunotherapy aimed at targeting HVEM reduced tumor growth twofold in vivo in a humanized mouse model of the pathology. The mode of action of the therapy was dependent on CD8^+^ T cells and is associated with improved T cell activation and reduced exhaustion. Finally, we demonstrated that HVEM expressed by the tumor negatively regulated the anti-tumor immune response. Our results indicate that targeting HVEM might be an attractive option for patients with prostate cancer.

**Abstract:**

The herpes virus entry mediator (HVEM) delivers a negative signal to T cells mainly through the B and T lymphocyte attenuator (BTLA) molecule. Thus, HVEM/BTLA may represent a novel immune checkpoint during an anti-tumor immune response. However, a formal demonstration that HVEM can represent a target for cancer immunotherapy is still lacking. Here, we first showed that HVEM and BTLA mRNA expression levels were associated with a worse progression-free interval in patients with prostate adenocarcinomas, indicating a detrimental role for the HVEM/BTLA immune checkpoint during prostate cancer progression. We then showed that administration of a monoclonal antibody to human HVEM resulted in a twofold reduction in the growth of a prostate cancer cell line in NOD.SCID.gc-null mice reconstituted with human T cells. Using CRISPR/Cas9, we showed that the therapeutic effect of the mAb depended on HVEM expression by the tumor, with no effect on graft vs. host disease or activation of human T cells in the spleen. In contrast, the proliferation and number of tumor-infiltrating leukocytes increased following treatment, and depletion of CD8^+^ T cells partly alleviated treatment’s efficacy. The expression of genes belonging to various T cell activation pathways was enriched in tumor-infiltrating leukocytes, whereas genes associated with immuno-suppressive pathways were decreased, possibly resulting in modifications of leukocyte adhesion and motility. Finally, we developed a simple in vivo assay in humanized mice to directly demonstrate that HVEM expressed by the tumor is an immune checkpoint for T cell-mediated tumor control. Our results show that targeting HVEM is a promising strategy for prostate cancer immunotherapy.

## 1. Introduction

Immune escape by tumors is now considered a hallmark of cancer [1]. Many immune mechanisms may explain the loss of tumor control, including defective MHC function and expression, recruitment of suppressive immune cells, and expression of co-inhibitory receptors such as PD-L1 [2]. In the last few years, targeting co-inhibitory receptors with monoclonal antibodies has shown impressive results in tumor regression and overall survival, leading to the approval of anti-CTLA-4, anti-PD-1, and anti-PD-L1 in numerous cancers [3]. However, the success of immune checkpoint inhibitors (ICI) remains partial as many patients fail to respond. This is particularly relevant for prostate cancer (PCa), the second deadliest cancer in the industrialized world and for which numerous clinical trials using ICI monotherapy have been disappointing [4]. Limited tumor infiltrates (cold tumors) or low expression of the targeted molecule may explain the relative inefficiency of ICI [5,6]. To overcome these limitations, other pathways that might be involved in immune escape must be investigated as these could complement actual therapies.

Recently, a new co-inhibitory pair was identified as a checkpoint of the anti-tumor immune response: HVEM (herpes virus entry mediator, TNFRSF14) and BTLA (B and T lymphocyte attenuator) [7]. Many immune cells can express these two molecules, including T and B cells, and BTLA is involved in the regulation of many biological processes [8]. HVEM signaling through BTLA inhibits T cell activation through recruitment of Src-homology domain 2-containing protein tyrosine phosphatases, SHP-1, and SHP-2 to the phosphorylated tyrosines of immunoreceptor tyrosine-based inhibitory motif [9]. Additionally, the HVEM network includes many additional partners, such as LIGHT, herpes simplex virus-1 (HSV-1), glycoprotein D (gD), lymphotoxin α (Ltα), or CD160 [7]. The binding of HVEM to CD160 is associated with the inhibition of CD4^+^ T cell activation [10], whereas it activates CD8^+^ T cells [11] and NK cells [12].

LIGHT is a T cell activator because transgenic expression of LIGHT in T cells leads to massive activation, especially in mucosal tissues [13]. In contrast, stimulation of HVEM expressed by T cells by any of its ligands is associated with proliferation, survival, and production of inflammatory cytokines, such as IL-2 and IFN-γ [14,15]. Thus, the HVEM network is complex, and T cell activation or inhibition may follow HVEM engagement depending on cis- or transactivation by various ligands in various types of cells.

Although the role of HVEM in T cell activation is well established, the role of HVEM expressed by tumor cells on the immune system is not clear. Several clinical studies have shown that HVEM expression is upregulated in many types of cancers including colorectal cancers [16], melanomas [17], esophageal carcinomas [18], gastric cancers [19], hepatocarcinomas [20], breast cancers [21], lymphomas [22], and PCa [23]. In these studies, a worse prognosis and lower survival of patients correlated with high levels of HVEM expression in the tumors or with elevated detection of soluble HVEM in the serum. Moreover, a reduction in the number of tumor-infiltrating leukocytes (TILs) is also associated with HVEM expression in tumors [13,15,17], indicating a detrimental role for HVEM in various cancers. In addition, different strategies to inhibit HVEM expression or function lead to increased T cell proliferation and function in syngeneic tumor mouse models [18,24,25]. However, to our knowledge, no study to date has assessed the possibility of using a monoclonal antibody (mAb) to HVEM to favor the anti-tumor immune response in a humanized context in vivo.

Similarly, upregulated expression of BTLA on CD8^+^ or CD4^+^ T cells or increased levels of soluble BTLA were associated with unfavorable prognosis for gallbladder cancer [26], diffuse large-B cell lymphoma [27], clear cell renal cell cancer [27], and prostate cancer patients [23]. Moreover, we have shown that tumor-specific CD8^+^ T cells are inhibited in melanoma patients through HVEM/BTLA signaling [28]. The safety of a mAb against BTLA is currently being evaluated in a clinical trial (NCT04278859).

To generate humanized mice, we used immuno-compromised NOD.SCID.γc-null (NSG) mice, which are deprived of murine T, B, and NK cells, but that retain functionally immature macrophages and granulocytes [29]. Patient-derived xenografts (PDX) have been shown to faithfully recapitulate tumor architecture and clinical features in breast [30], ovarian [31], lung [32], skin [33], and prostate [34] cancers. However, it is challenging to generate PDX mouse models on a regular and consistent basis. Humanized mice grafted with human cancer cell lines have been used in hundreds of studies to evaluate the efficacy of various treatments for cancer immunotherapy [35]. Regardless of the mode of reconstitution of the human immune system in NSG mice (from progenitors or peripheral blood, as herein), all suffer from the same drawback that human T cells will be allogeneic to the tumor. This would prevent the tumor antigen-specific response to be accurately measured in humanized mice. Nevertheless, accumulating evidence indicates that, in addition to their different proportions in the naïve T cell repertoire, a high degree of similarity exists between antigen-specific and allogeneic-specific immune responses [36,37]. Furthermore, T cell response to the tumor is not restricted to antigen-specific T cells but can also imply bystander T cells, which are not specific but actively engaged in tumor control [38,39]. Thus, despite some limitations, humanized mice may still provide valuable clues on therapeutic strategies aimed at enhancing the allogeneic immune response mediated by human T cells to the tumor, with relevance to T cell-mediated tumor antigen-specific and bystander immune responses in cancer patients. In this study, we investigated the therapeutic potential of a monoclonal antibody targeting human HVEM in humanized mice and the underlying cellular and molecular mechanisms associated with therapeutic efficacy.

## 2. Results

### 2.1. HVEM and BTLA Were Associated with Lower Progression-Free Intervals in Prostate Cancer Patients

Analysis of TCGA (The Cancer Genome Atlas) revealed that HVEM mRNA was expressed at high levels in the PRAD (PRostate ADenocarcinomas) dataset (normalized log_2_ counts > 10) and higher levels in primary tumors than in patient-matched normal tissues (Figure 1A). In contrast, BTLA was expressed at low (normalized log_2_ counts < 3) and similar levels in the tumor and patient-matched normal tissues (Figure 1B). For patients with Gleason scores of 6 or 7 (low-grade adenocarcinomas), above-median expression of HVEM was associated with a lower progression-free interval (PFI) over 5 years, whereas this was not observed for BTLA (Figure 1C). In contrast, HVEM mRNA expression levels were not correlated with good or poor PFI in patients with more advanced disease (Gleason scores of 8 or 9), but above-median expression of BTLA was associated with a lower PFI (Figure 1D). Overall, the above-median expression of HVEM and BTLA together was a poor prognostic factor for PFI irrespective of the clinical score (Figure 1E). Interestingly, this was not observed for the PD1/PD-L1 co-inhibitory signaling pathway (Figure 1F). These results indicate a detrimental role of HVEM/BTLA in PCa at all stages of the disease and suggest that blocking the HVEM/BTLA pathway might be an attractive option for PCa patients. Accordingly, we focused the rest of the study on PCa cell lines in vivo.

### 2.2. Targeting HVEM with a mAb Improved Tumor Control in Humanized Mice

We first determined whether targeting HVEM with a mAb could impact tumor growth in vivo. For this, we implanted prostate cancer cell lines in NSG mice and grafted human PBMCs a few days later. No differences in tumor growth were observed in mice grafted with the prostate cancer cell line DU145, which did not express HVEM (Figure 2A). In contrast, a twofold reduction in tumor growth was observed in mice grafted with the HVEM-positive PC3 prostate cancer cell line (Figure 2B). The presence of human cells was mandatory for the efficacy of the mAb since tumor growth was unaffected by the treatment in non-humanized NSG mice (Appendix A). The lack of an effect in non-humanized mice also indicates that the mAb did not directly kill the PC3 cell line in vivo. To formally demonstrate that HVEM on the tumor was required for therapeutic efficacy of the mAb, we generated an HVEM-deficient PC3 cell line (clone 1B11) using CRISPR-Cas9 ribonucleoprotein (RNP) transfection. Treatment with the mAb was completely inefficient on the 1B11 clone in humanized mice (Figure 2C), showing that HVEM expression by the tumor was mandatory for the therapeutic efficacy of the mAb. Notably, the mAb was also effective at controlling tumor growth in humanized mice grafted with the HVEM-positive melanoma cell line Gerlach but not with the HVEM-negative triple-negative breast cancer (TNBC) cell line MDA-MB-231 (Appendix A). Altogether, these results indicate that the therapeutic efficacy of targeting HVEM was independent of the tumor tissue origin but rather strictly dependent on HVEM expression by the tumor.

### 2.3. Tumor Control Was Dependent on CD8^+^ T Cells in Anti-HVEM-Treated Mice

To investigate the mode of action of the mAb, we determined the composition of human CD45^+^ cells in the PC3 tumor and the spleens of treated and untreated animals by flow cytometry. To determine the efficiency of human cell reconstitution independent of technical variability linked to sample preparation, we first established the frequencies of human CD45^+^ cells relative to the added frequencies of murine and human CD45^+^ cells (Appendix A). Human CD45^+^ cells represented close to 100% of all CD45^+^ cells in the spleen (Appendix A). Among hCD45^+^ cells, CD3^+^ T cells represented 90% (Appendix A), in which CD4^+^ and CD8^+^ cells represented 60% and 30%, respectively (Appendix A). Likewise, human CD45^+^ cells represented 90% of all CD45^+^ cells in the tumor (Appendix A). Among hCD45^+^ cells, CD3^+^ T cells represented more than 98% (Appendix A), in which CD4^+^ and CD8^+^ cells represented 40% and 60%, respectively (Appendix A). We noticed that the CD4 to CD8 ratio was inverted in the tumor relative to the spleen (Appendix A), indicating that CD8^+^ T cells might have a preferential tropism to the PC3 tumor in humanized mice, as previously described [40]. Importantly, treatment did not affect any of these proportions. In contrast, we observed an increase in the frequencies of both CD4^+^ and CD8^+^ T cells expressing the proliferation marker Ki67 (Figure 3A,B). Additionally, CD8 but not CD4 T cell numbers (represented as z-scores to normalize the numbers across experiments) were significantly increased in the anti-HVEM-treated group relative to the control group (Figure 3C), suggesting a role for CD8^+^ T cells in tumor control. However, the frequencies of CD8^+^ and CD4^+^ T cells expressing the cytotoxic molecules granzyme B or perforin were not elevated in the tumor or spleen of the anti-HVEM group (Appendix A). To directly determine the contribution of CD8^+^ T cells to tumor control in anti-HVEM-treated mice, we investigated tumor growth in mice depleted of human CD8^+^ T cells. Depletion of CD8^+^ T cells was efficient in most of the mice analyzed, as assessed by the frequencies of CD4^−^CD3^+^ cells remaining after treatment (Appendix A). Depletion of CD8^+^ T cells reversed the effect of the anti-HVEM mAb (Figure 3D), showing that tumor control was dependent on CD8^+^ T cells.

### 2.4. Treatment with the Anti-HVEM mAb Did Not Increase Graft vs. Host Disease Nor the Number or Proliferation of Human T Cells

Our observations thus far are consistent with the hypothesis that targeting HVEM on the tumor releases an inhibitory signal on CD8^+^ T cells, allowing their proliferation in situ. We wanted to rule out the possibility that the mAb behaves as an agonist, directly activating HVEM^+^ CD8^+^ T cells in vivo, leading to better tumor control. One prediction of this hypothesis would be that GVHD occurring in NSG mice grafted with the tumor and reconstituted with human T cells would be exacerbated following anti-HVEM administration. However, despite the impact of the mAb on tumor growth (Figure 2), we observed similar weight loss and mortality in anti-HVEM or isotype control-treated mice (Figure 4A,B). Furthermore, in contrast to TILs, the activation of human T cells in the spleens was the same in both groups, as judged by similar numbers of CD4^+^ and CD8^+^ T cells and similar frequencies of Ki67^+^ cells (Figure 4C,D). Thus, anti-HVEM therapy in humanized mice reduced the growth of HVEM^+^ tumors by a mechanism that was independent of a systemic agonist effect of the mAb on human T cells.

### 2.5. mRNA Enrichment Analysis Showed Increased Activation and Decreased Immuno-Suppression in TILs of Anti-HVEM-Treated Mice

To better characterize the anti-tumor immune response following mAb treatment, we established a list of differentially expressed genes (DEG) in sorted hCD45^+^ TILs from mice treated with the anti-HVEM mAb or its isotype control. For the analysis, 287 genes with a raw count higher than 55 and an absolute fold-change of at least 20% were set to be differentially expressed. Among the 287 genes, 145 were upregulated with a log_2_FC > 0.26, and 142 were downregulated (log_2_FC < −0.3) in HVEM-treated mice relative to isotype-treated controls (Figure 5A). Several interleukins and chemokines genes signing T cell activation were enriched in the treated group, such as *LTA*, *IL22*, *IL32*, *CCL5*, and *CCL4*. Of note, *GZMB* and *PRF1* were among the genes with the highest levels of expression, but the difference between the groups was weak, confirming our observation by flow cytometry (Appendix A). Gene set enrichment analysis (GSEA) identified the “JAK-STAT signaling pathway” signature as significantly and positively enriched in TILs of HVEM-treated mice (Figure 5B). In addition, upregulated genes in the anti-HVEM group were enriched in members of several ontologies related to lymphocyte activation, including the tumor necrosis factor-mediated signaling pathway, cytokine and chemokine binding and activity, and T cell receptor complex (Appendix A). On the other hand, some genes belonging to immunosuppressive pathways were downregulated in HVEM-treated TILs such as *ENTPD1* (CD39); *IL10*; and the co-inhibitory receptors *BTLA*, *TIGIT*, *LAG3*, and *HAVCR2* (TIM3), as well as the “don’t eat me” receptor *CD47* (Figure 5A). We verified by flow cytometry that BTLA was indeed down modulated on human T cells following anti-HVEM treatment (Appendix A). In addition, GSEA showed that the “immunoregulatory interactions between a lymphoid and a non-lymphoid cell” signature was significantly repressed in the DEG signature (Figure 5C), signing altered adhesion and motility of TILs. Overall, anti-HVEM treatment was associated with profound modifications of TILs, with increased expression of genes belonging to activation and proliferation signaling pathways and decreased expression of genes signing an exhausted phenotype.

### 2.6. HVEM Was an Immune Checkpoint during Anti-Tumor T Cell Immune Response in Humanized Mice

A reduction in tumor growth in the absence of HVEM was already apparent for the isotype-treated group, as shown in Figure 2B,C. Thus, we directly compared the growth of HVEM-positive or HVEM-negative PC3 cells in NSG mice with or without the addition of human PBMCs (Figure 6). Both cell lines grew equally well in non-humanized NSG mice (Figure 6A), showing that HVEM deficiency did not affect in vivo tumor development per se. In contrast, tumor growth of the 1B11 clone was reduced twofold compared to that of the parental PC3 cell line in humanized mice (Figure 6B). Furthermore, T cell proliferation was significantly increased in the absence of HVEM from the tumor (Figure 6C). Thus, removing HVEM from the tumor released a brake on the allogeneic T cell response to the tumor, demonstrating that HVEM was an immune checkpoint under these experimental conditions.

## 3. Discussion

Here, we report improved tumor control by human T cells in vivo following administration of a mAb to HVEM. Moreover, we deciphered the mode of action of the mAb in vivo using complementary technologies. Furthermore, we propose a simple in vivo assay for immune checkpoint discovery and validation. To our knowledge, the present report is the first to combine CRISPR/Cas9-mediated deletion of putative checkpoints with the assessment of tumor growth in humanized mice. One limitation of the assay is that PBMC-humanized mice are mostly reconstituted with T cells, as shown herein, limiting the usefulness of the assay to T cell-specific immune checkpoints. Another limitation is the allogeneic nature of the immune response to the tumor in humanized mice, which would only be circumvented in PDX models with autologous human immune cells, an ongoing effort but still challenging to set up [41]. Despite these limitations, we believe that our in vivo assay will be of great help in investigating other candidates in more advanced models of humanized mice, that is, mice reconstituted with human hematopoietic progenitors.

We show that the HVEM/BTLA checkpoint could be exploited for therapy in humanized mice using a mAb to human HVEM. We found that HVEM expression by the tumor was necessary and sufficient to elicit tumor control by the mAb since it had no effect on HVEM-negative cell lines and no agonist activity on human T cells. Park et al. showed in a syngeneic mouse model that transfecting an agonist scFv anti-HVEM in tumor cells resulted in increased T cell proliferation, improved IFN-γ and IL-2 production and improved tumor control [24]. In addition to the species differences, the discrepancy with our results could be explained by the fact that T cells are strongly activated in huPBMC mice [42]. The downregulation of HVEM expression upon activation [43] may have limited the binding of the anti-HVEM antibody to T cells in our model. Thus, it remains possible that the mAb would behave differently in humans. In contrast, BTLA is upregulated upon T cell activation [44], increasing the susceptibility of T cells to inhibition by HVEM^+^ tumor cells [16,18,20,45]. The opposite was observed in the tumor microenvironment following treatment, with an increase in *HVEM* and a reduction of *BTLA* gene and protein expression, with a concomitant increase in *LTA* and *LIGHT*, two other ligands for HVEM. It is important to note that the binding sites of LIGHT and BTLA differ on HVEM [46]. Thus, the anti-HVEM mAb might have limited inhibition of activated T cells through blockade of HVEM binding with BTLA but not with the other ligands that are T cell activators. An alternative possibility would be that LIGHT and LTα in their soluble forms inhibit the interaction of HVEM with BTLA [47]. As of today, reciprocal regulation of HVEM and BTLA has not been reported but our observation is reminiscent of earlier findings showing reciprocal regulation of HVEM by LIGHT [43].

Previous studies in mice have also shown that inhibiting HVEM expression on the tumor or its interaction with its ligands has a positive effect on T cells. Injection of a plasmid encoding a soluble form of BTLA (to compete with endogenous BTLA for HVEM) was associated with an increase in inflammatory cytokine production by TILs and a decrease in anti-inflammatory cytokines at the RNA level [25]. Similarly, vaccination with a tumor-associated antigen was more efficient if HVEM interactions with its ligands were blocked by HSV-1 gD, allowing regression of a large tumor mass [48]. Moreover, silencing HVEM expression in the tumors with siRNA was also associated with an increase in CD8 T cells and inflammatory cytokine production in a murine colon carcinoma model [18]. In addition, the use of siRNA to HVEM on ovarian cancer in vitro promoted T cell proliferation and TNF-α and IFN-γ production [49]. Numerous results from our study also support increased T cell activation in the absence of HVEM/BTLA signaling: (i) TILs from mice treated with anti-HVEM were enriched in ontologies signing activation by cytokines, chemokines, and signaling pathways that are well-known inducers of proliferation, differentiation, migration, and apoptosis; (ii) comparison between TILs from mice treated with the anti-HVEM or isotype control mAb also highlighted decreased expression of many co-inhibitory receptor genes (BTLA, TIGIT, LAG3, and HAVCR2 [50,51]) or with immunosuppressive functions (ENTPD1 and IL10), suggesting a lower exhaustion status; and (iii) TILs from anti-HVEM-treated mice or the HVEM-null PC3 cell line had increased frequencies of Ki67^+^ cells relative to controls, signing improved proliferation. Altogether, these results strongly suggest that HVEM expressed by the tumor negatively regulated T cell-mediated control of tumor growth.

Interestingly, we observed that CD8^+^ T cell depletion in vivo favored the growth of PC3 tumors in anti-HVEM-treated mice. It might be inferred from this result that CD8^+^ T cells would also play a role in the absence of the treatment. However, a major role for human CD4^+^ T cells in PC3 tumor control in NSG mice has been reported [40]. In this study, CD4^+^ or CD8^+^ T cells were depleted before injection into mice. Despite depletion, a significant proportion of CD8 T cells were still present in the tumor although they were absent from the spleen, casting doubt on the interpretation of these results. It would be interesting to evaluate the role of CD4^+^ and CD8^+^ T cells in the absence of treatment in our model. Notably, a similar implication of CD8^+^ T cells on tumor control in humanized mice was reported for the TNBC MDA cell line treated with the anti-PD1 pembrolizumab [52]. Overall, we propose a model in which treatment with the anti-HVEM mAb would block the HVEM/BTLA inhibitory signaling on CD8^+^ TILs that would increase their proliferation and numbers, associated with a reduction of their exhausted phenotype, ultimately leading to better tumor control.

## 4. Conclusions

The recent success of ICI for cancer immunotherapy (anti-CTLA-4, anti-PD-1/PD-L1) has confirmed the hypothesis that the immune system can control many cancers, but disappointing results were obtained for PCa [4] which is in line with our observation that PD1/PD-L1 is not associated with lower PFI in PCa patients. In light of the promising results reported herein, anti-HVEM therapy might be combined with ICI and/or chemotherapy to further enhance anti-tumor immunity in PCa.

## 5. Materials and Methods

### 5.1. Preparation of Human Peripheral Mononuclear Cells

Human peripheral blood was obtained from Etablissement Francais du Sang (EFS) after obtaining informed consent from the donor. Human peripheral blood mononuclear cells (PBMC) were isolated using a Biocoll density gradient (Biochrom GmbH, Berlin, Germany). Cells were washed in PBS 3% FCS and diluted to the appropriate concentration in 1× PBS before injection into mice.

### 5.2. Humanized Mice Tumor Model

All animals used were NSG mice (stock ≠ 005557) purchased from the Jackson Laboratory (Bar Harbor, ME, USA). To assess therapeutic activity, 8–20-week-old males and females NSG mice were injected subcutaneously with 2 × 10^6^ tumor cells. One week later, mice were irradiated (2 Gy) and grafted the same day with 2 × 10^6^ huPBMC by retro-orbital injection. This experimental protocol was inspired by our previous work on setting up the lower limit of grafted PBMCs to limit GVHD in NSG mice [53]. Four to five days after transplantation, the anti-huHVEM antibody or isotype control was injected intraperitoneally at 2 mg/kg. General health status, body weight, and survival of mice were monitored every 3–4 days to evaluate GVHD progression. Mice with overt signs of GVHD, such as weight loss greater than 20% of their initial weight, hunched back, ruffled fur, and reduced mobility, were immediately sacrificed. For CD8 depletion, the optimal conditions and efficacy assessments have been previously described [54]. Briefly, mice were injected intraperitoneally with 10 mg/kg of the anti-CD8 MT807R1 (rhesus recombinant IgG1 provided by the Nonhuman Primate Reagent Resource [55]) or the isotype control (clone DSPR1) the day following humanization.

### 5.3. Antibodies

The clone 18.10 has been described previously [56]. Briefly, 18.10, a murine IgG1 anti-human HVEM mAb, was produced as ascites and purified by protein A binding and elution with the Affi-gel Protein A MAPS II Kit (Bio-Rad, Marnes-La-Coquette, France). A Mouse IgG1 isotype control (clone MOPC-21 clone) was purchased from BioXCell (West Lebanon, NH, USA).

### 5.4. Cell Lines

PC3 (non-hormonal-dependent human prostate cancer cells) and DU145 (prostate cancer cells) were grown in high-glucose DMEM supplemented with 10% FCS, L-glutamine, and antibiotics (penicillin/streptomycin, ThermoFisher, Les Ulis, France). The PC3 cell line was genetically authenticated before the initiation of the experiments (Eurofins). All cells were confirmed to be free of mycoplasmas before injection into mice using the MycoAlert detection kit (Lonza, Basel, Switzerland). Tumor growth was monitored using an electronic caliper, and volumes were determined using the following formula: [(length × width^2^)/2]. The animals were sacrificed and analyzed 21 days after the humanization.

### 5.5. Generation of HVEM-Deficient PC3 Clone Using CRISPR-Cas9 RNP Transfection

A total of 50,000 PC3 cells were seeded into a 24-well plate. Twenty-four hours later, cells were incubated with sgRNA complementary to exon 3 of HVEM (GCCAUUGAGGUGGGCAAUGU + Scaffold, TrueGuide synthetic guide RNAs, Invitrogen™, ThermoFisher), Cas9 nuclease (TrueCut™ Cas9 Protein v.2, Invitrogen™, ThermoFisher), and Lipofectamine (Lipofectamine™ CRISPRMAX™ Cas9 Transfection Reagent, Invitrogen™, ThermoFisher) according to the manufacturer instructions (TrueCut Cas9 Protein v.2 (27/09/2017)). After three days, the efficiency was evaluated using the GeneArt Genomic Cleavage Detection Kit (Invitrogen™, ThermoFisher) according to the manufacturer’s instructions. For this assay, DNA was amplified using the following primers: TGCGAAGTTCCCACTCTCTG (forward) and GGATAAGGGTCAGTCGCCAA (reverse). The cells were cloned by limiting dilution in 96-well plates. Clones were screened for HVEM expression by flow cytometry using anti-HVEM (clone 94801, BD, Le Pont de Claix, France) and were considered negative if HVEM expression was undetectable for at least three subsequent measurements.

### 5.6. Phenotypic Analysis by Flow Cytometry

Tumors were digested with 0.84 mg/mL collagenase IV and 10 μg/mL DNAse I (Merck KGaA, Darmstadt, Germany) for 40 min at 37 °C with intermediate flushing of the tissue. Cells were passed through a 100 µm cell strainer and suspended in PBS containing 3% FCS. To eliminate dead cells and debris, we isolated tumor cell suspensions using a Biocoll gradient. Rings were collected and washed, and cell pellets were suspended in PBS 3% FCS before counting on the LUNA™ Automated Cell Counter (Logos Biosystems, Villeneuve d’Ascq, France). Subsequently, up to 2 × 10^6^ live cells were stained with a viability dye (eF506, Fixable Viability Dye, ThermoFisher) for 12 min at 4 °C. Fc receptor binding was blocked with human FcR Blocking Reagent (120-000-442, Miltenyi Biotec, Paris, France) and anti-CD16/32 (clone 2.4G2) for 10 min. The following antibodies were added for 35 min at 4 °C: hCD45-BUV805 (HI30, BD), hCD3-PECyn7 (SK7, BD), hCD4-PerCP (RPA-T4, BioLegend, Amsterdam, The Netherlands), hCD8-APC-H7 (SK1, BD), hKi67-AF700 (B56, BD), hCD270-BV421 (cw10, BD), hBTLA-PE-CF594 (J168-540, BD), mCD45-BUV395 (30-F11, BD), hGranzymeB-APC (GB11, eBioscience, ThermoFisher), and hPerforin-PE (B-D48, BioLegend). For intracellular staining, Foxp3/transcription factor staining (eBioscience, ThermoFisher) or Cytofix/Cytoperm (BD) buffer sets were used. Cells were washed with 1× PBS before acquisition on an ×20 cytometer (BD). The absolute count of different populations was determined by adding 50 µL of cell counting beads (Bangs Laboratories Fishers, IN, USA) before acquisition. Data were analyzed using FlowJo software (TreeStar, Ashland, OR, USA).

### 5.7. NanoString nCounter Expression Assay

For the NanoString^®^ experiment, 14- to 15-week-old NSG mice were humanized and treated with anti-HVEM or isotype. On day 28 post-humanization, tumors were harvested and TILs were isolated as described above. To maximize mRNA recovery, we pooled TILs by treatment groups (four mice in the anti-HVEM group and five in the isotype control group). Cells were then stained with a viability dye (eF506) and anti-hCD45-APC (HI30, BioLegend). Live hCD45^+^ cells were sorted using an Aria II cell sorter. After centrifugation, the cells were suspended in RLT buffer (Qiagen, Les Ulis, France) before freezing at −80 °C until analysis. Data were normalized using NanoString’s intrinsic negative and positive controls according to the normalization approach of the nSolver analysis software (Nanostring, Seattle, WA, USA).

### 5.8. Bioinformatics Analysis

For ontology enrichment analysis, only genes upregulated by the treatment were analyzed using the enrichment analysis visualization Appyter to visualize Enrichr results [57]. The list of DEG (up- and downregulated) was ranked by fold-change for pre-ranked GSEA. Enrichment was performed with the C2 Canonical Pathways v.7.4 gene set using the GSEA 4.1.0 Linux desktop application [58] from the Broad Institute. With that workflow, a false discovery rate (FDR) or a family-wise error rate (FWER) less than 0.25 is deemed “significant”. The Cancer Genome Atlas (TCGA) database was analyzed using the Xena browser (http://xena.ucsc.edu, accessed on 9 June 2021) provided by the University of California (Santa Cruz, CA, USA) [59]. The Prostate Adenocarcinoma (PRAD) dataset was used with subsequent filterings on TNFRSF14, BTLA, PDCD1, and CD274 mRNA expression levels and Gleason clinical scores.

### 5.9. Statistical Analysis

All statistical tests were performed using Prism v.8 (GraphPad Inc., La Jolla, CA, USA) or JASP v.0.14.3 (available at https://jasp-stats.org, accessed on 9 June 2021). The nature of the statistical test used to compare results is indicated in each figure legend. When necessary, the p-values of these tests are indicated in the figure panels. The statistical power of the analyses (alpha) was set arbitrarily at 0.05.

## Figures and Tables

**Figure 1 cancers-13-03009-f001:**
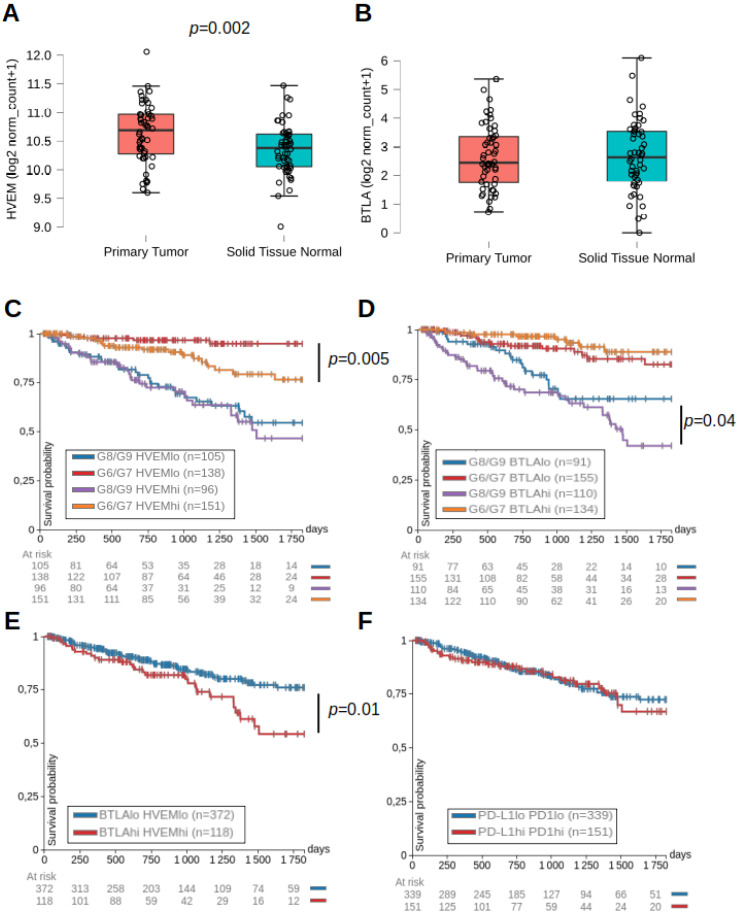
HVEM and BTLA were associated with lower progression-free intervals in prostate cancer patients. Expression of HVEM (**A**) and BTLA (**B**) mRNA (expressed as log_2_ of normalized counts) in matched patients in the primary tumor or normal tissue (*n* = 52). The *p*-value indicated on the graph is from a paired Student’s *t*-test with normality assumption validated by a Shapiro–Wilk test. Data were extracted from the PRAD dataset of the TCGA using Xena. Progression-free intervals (PFI) of PCa patients according to the level of expression of HVEM (**C**) or BTLA (**D**) and the Gleason scores signing low-grade (G6/G7) or high-grade (G8/G9) adenocarcinomas over a 5-year follow-up. For each group of clinical scores, patients were split into two groups according to below-median (HVEM^lo^ or BTLA^lo^) or above-median (HVEM^hi^ or BTLA^hi^) mRNA expression. PFI curves of PCa patients harboring below-median or above-median expression of both HVEM and BTLA (**E**) or both PD1 and PD-L1 (**F**) independently of clinical scores. The number of patients in each group is indicated in brackets and the number of patients at risk at main time points is indicated below the graph. The *p*-values on the graphs are from a log-rank test in Xena.

**Figure 2 cancers-13-03009-f002:**
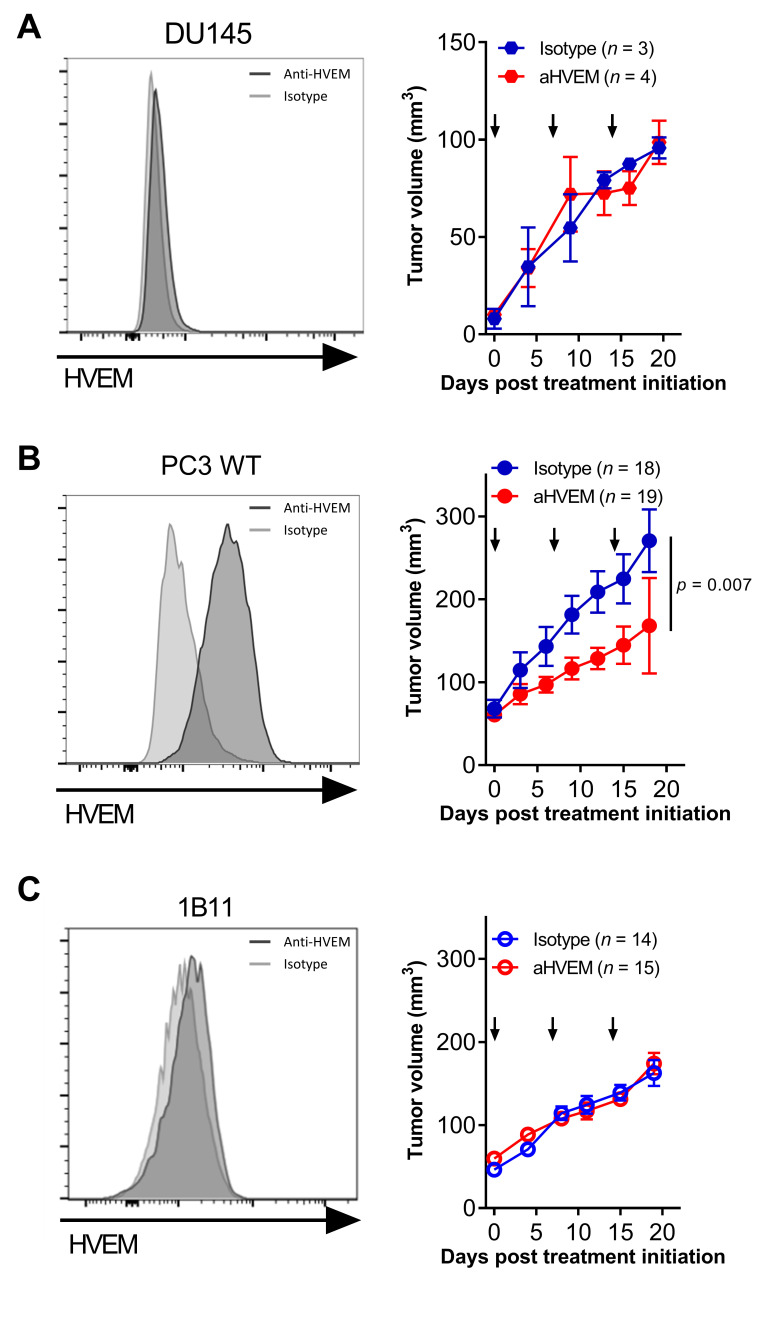
Targeting HVEM with a mAb improved tumor control in humanized mice. HVEM expression and tumor growth of the prostate cancer cell lines DU145 (**A**), PC3 (**B**), and the HVEM-deficient PC3 clone 1B11 (**C**). HVEM expression was determined by flow cytometry with the anti-HVEM mAb (clone 18.10) and a secondary antibody. Curves represent the cumulative mean tumor volume (±SEM) from one experiment with DU145, two for 1B11, and three for PC3. The number of mice at the beginning of the experiments is indicated in brackets. Arrows indicate the time of the injections. The *p*-value reported on the graph tests the null hypothesis that the slopes are identical using a linear regression model.

**Figure 3 cancers-13-03009-f003:**
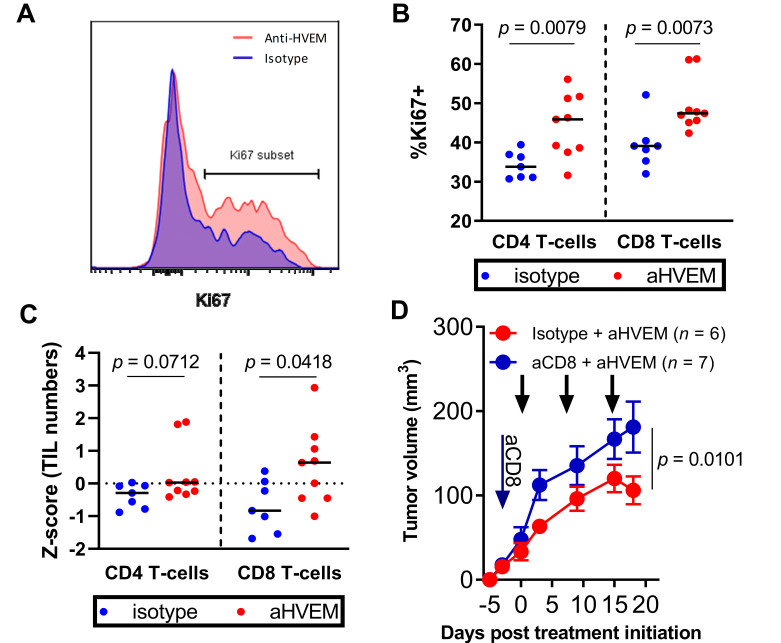
Tumor control was dependent on CD8^+^ T cells. (**A**) Representative histogram of Ki67 staining. (**B**) Frequencies of Ki67-expressing cells among CD4^+^ and CD8^+^ T cells in the tumor. Data are cumulative of two independent experiments performed at D21 post-humanization. (**C**) Absolute numbers of CD4^+^ and CD8^+^ T cells in PC3 tumors from two independent experiments. To allow comparison, we converted numbers obtained in each mouse to z-scores (number of standard errors separating the samples from the mean). Each dot is a mouse. The *p*-value on the graphs is the result of a Mann–Whitney non-parametric *t*-test. (**D**) Growth of the PC3 cell line in humanized mice treated with anti-HVEM mAb and depleted or not of their CD8 T cells. CD8 T cells were depleted on the day following humanization (blue arrow). Curves are the mean tumor volume (±SEM) in the indicated number of mice. Black arrows indicate the time of anti-HVEM mAb injection. Data are cumulative of two independent experiments. The *p*-value reported on the graphs tests the null hypothesis that the slopes are all identical using a linear regression model.

**Figure 4 cancers-13-03009-f004:**
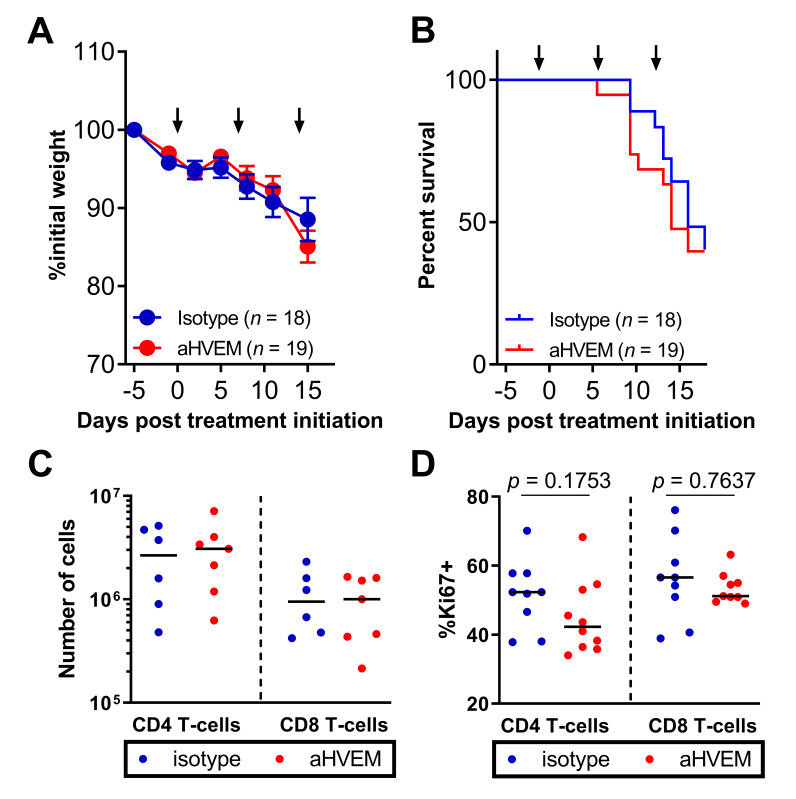
Treatment with the anti-HVEM mAb did not increase GVHD nor the number or proliferation of human T cells in the spleen. Percentages of initial weight (**A**) and survival (**B**) of tumor-bearing humanized NSG mice following treatment by the anti-HVEM mAb or an isotype control are shown. Data shown are cumulative of 3 independent experiments. Numbers (**C**) and frequencies of Ki67^+^ cells (**D**) at D21 post humanization in the indicated subsets and are cumulative of two independent experiments. Each dot is a mouse. The *p*-values are from a Mann–Whitney *t*-test.

**Figure 5 cancers-13-03009-f005:**
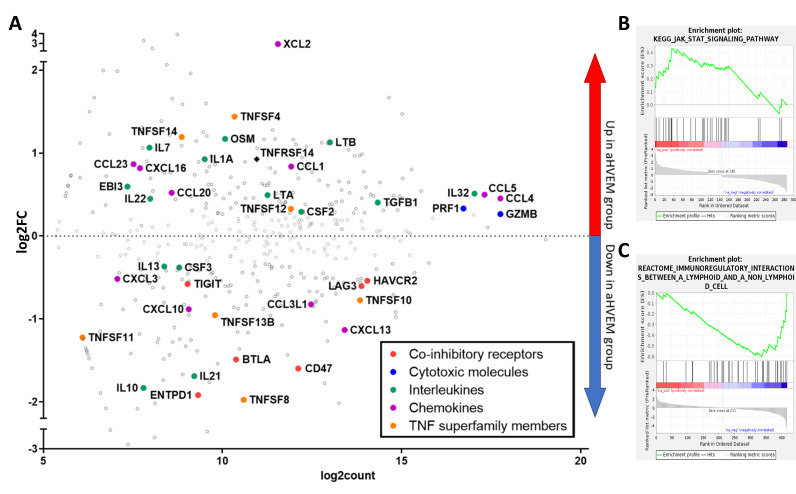
mRNA enrichment analysis showed increased activation and decreased immunosuppression in TILs of anti-HVEM treated mice. (**A**) MA-plot comparing gene expression between TILs from anti-HVEM and isotype-treated mice. Represented are the log_2_ fold-change in the expression of a given gene between anti-HVEM or isotype-treated mice (log_2_FC, *y*-axis) vs. the mean absolute count after normalization (log_2_count). Only genes with raw counts higher than 55 are shown. Some notable genes were manually annotated according to their biological functions by the indicated color code. (**B**) GSEA identified the “JAK-STAT signaling pathway” signature as significantly enriched (normalized enrichment score (NES = 1.81, *p*-value = 0.004, q-value = 0.17 (FDR), *p*-value = 0.32 (FWER)) in genes upregulated by the treatment. (**C**) “Immunoregulatory interaction between a lymphoid and a non-lymphoid cell” signature was significantly enriched (NES = −1.86, *p*-value = 0.001, q-value = 0.15 (FDR), *p*-value = 0.25 (FWER)) in genes downmodulated by the treatment.

**Figure 6 cancers-13-03009-f006:**
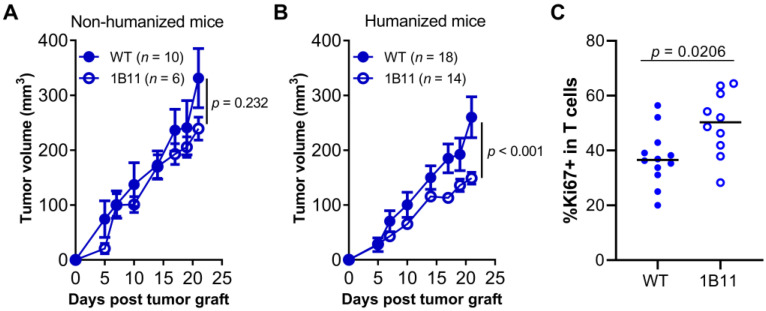
HVEM was an immune checkpoint during anti-tumor T cell immune response in humanized mice. Growth of the indicated PC3 cell lines (WT or 1B11) in non-humanized (**A**) or PBMC-humanized mice (**B**). Curves are the mean tumor volume (±SEM) in the indicated number of mice. Data are cumulative of at least two experiments. The *p*-value on the graphs tests the null hypothesis that the slopes are equal using a linear regression model. (**C**) Frequencies of Ki67^+^ in CD4^+^ and CD8^+^ T cells in PC3 WT or 1B11 tumor at day 21 post-humanization from one experiment with 6 and 5 mice, respectively. The *p*-values are from a Mann–Whitney *t*-test.

## Data Availability

The data presented in this study are available upon request from the corresponding author.

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
