# Peer review of "Blockade of HVEM for Prostate Cancer Immunotherapy in Humanized Mice"

_cancers, 2021, doi:10.3390/cancers13123009_

Round 1

Reviewer 1 Report

In this manuscript, Aubert N. reported that anti-HVEM mAb may be a potential effective therapy in PCa humanized mouse model. By flow cytometry analysis, they showed that tumor expression of HVEM and CD8+T cells are the key players in the in vivo model. It is interesting to determine the anti-HVEM in humanized model, and overall the manuscript is informative. However, some of the conclusions are not well-supported by the current data. Meanwhile, some of experimental settings need to be improved.

Major comments:

  1. As the manuscript was mainly focused on HVEM/BTLA4 pathway, a literature review on their roles in regulating the anti-tumor T cell responses, at least in PCa need to be further explored in the introduction part.
  2. For the humanized mouse model, the engulfment of T cells in each mice need to be verified by flow cytometry. Meanwhile, how about the successful rate of mouse humanization? In addition, as the authors did irradiation (2 Gy) in mice after tumor injection, do they determine the impact of irradiation on tumor cells? Normally, people do humanization followed by tumor injection in humanized mouse models.
  3. Representative flow dot plots are needed in fig. 3. Meanwhile, to see the T cell distribution and proportions in an original spatial manner, IHC/IF on tumor slides are also necessary.
  4. Fig. 3D: how about the depletion efficacy? A data on CD8+T cell proportion after depletion is needed. Meanwhile, a control group with only CD8+T cell depletion is also missing here.
  5. As the authors already demonstrated that TILs were mainly T cells in fig. 3, the description of the title in 2.5 is not clear. It should only refer to T cell intrinsic mRNA analysis.
  6. What is the difference between Fig. 2B-C isotype treatment data and fig. 6B? It seems to be redundant data here. In addition, the comparison of tumor growth between two mouse models did not lead to the conclusion that ‘HVEM is an immune checkpoint during anti-tumor T cell immune response’.
  7. The authors indicated the importance of HVEM/BTLA checkpoint in PCa in introduction/conclusion. So, what is the role of BTLA? As a T cell immune checkpoint, the interaction between HVEM and BTLA is important for T cell function. Therefore, how about the expression levels of BTLA in T cells in vivo? Besides mRNA, protein level determined by flow cytometry or co-IF are needed. In addition, the functional assays of T cell activity in controlling by HVEM/BTLA are also needed to demonstrate the potential immune checkpoint roles.

Minor comments:

  1. The resolution of the figures need to be improved.
  2. Please label the lines in the overlay histogram in fig. 2.

Reviewer 2 Report

In this paper, Aubert et al, describe the unexplored role of HVEM as immune checkpoint.

HVEM delivers a negative role to T cells through the interaction with BTLA but, in parallel, is a T cell activator after LIGHT interaction. But HVEM can be also expressed in tumours, so this complex network needs further studies and any clue added will have an extremely relevance on cancer immunology.

The authors rely on a novel experimental design in humanized mice. They are able to demonstrate in vivo that targeting HVEM has the capacity to reduce tumour growth on HVEM expressing prostate cancer cell. They also provide the mechanisms behind mode of action of the antibody anti-HVEM: (i) at cellular level, dissecting the role of T lymphocytes within the tumour, and also (ii) at molecular level by the mRNA analysis.  Finally, they importantly conclude that this mechanism may be common for other tumours.

The results clearly justify the initial hypothesis to consider HVEM as a bona fide immune checkpoint.

Minor changes:

  • Point 5.2 from Material and Methods: Although it is indicated in the graphs, the authors should specify the days and the number of injections of the anti-HVEM antibody. In this line, in Figures 2 and 3, the X axis is related to “days post treatment initiation” referring to the first antibody injection. But for figure 4, the X axis corresponds to “days post-humanization”. My suggestion would be to follow the same criteria and, also,  it could be helpful to add a small outline for the protocol: day 0 tumor injection, day 7 PBMC graft, days 12, 19 and 26 antibody treatments.
  • Point 5.6 from Material and Methods. Please, indicate the day when animals are sacrificed and tumour analysed.
  • The immunomodulatory functions of HVEM/BTLA on T regulatory cells have been widely described. Have the authors studied the presence and activity of this T cell population within the tumors? This can be easily done by FACS or checking the GSEA result for FOXP3.
  • Figure 4B, despite the impact of anti-HVEM on tumour growth (Figure 2B), there is no difference on survival. The survival curves seem to equalize just after the last dose of antibody. Have the authors planned to increase the number of doses of antibody to force differences or they consider graft is no longer viable from day 20?
  • Figure 3B, CD4+ cells proliferate significantly more after anti-HVEM treatment as CD8+ do. Although in general terms tumor control is dependent on CD8 cells, the antitumor effect of CD4 cannot be ignored. They produce effector cytokines such as IFNγ and TNFα, which have direct anti-tumour activity and, further, CD4 cells can mediate direct cytotoxicity against tumour cells in a similar manner to CD8 under specific conditions, in both mouse tumour models and in patient-derived CD4.  It may be worth taking a look at CD4+ in this humanized model.

Round 2

Reviewer 1 Report

Necessary control groups are needed in the overall experimental setting, even with good reference support. Meanwhile, I am not quite understand the relationship of 'patent issue' and in vitro functional assay. Although it is interesting to see the anti-tumor effect of HVEM, the mechanism is still not clearly linking to the conclusion that HVEM is an immune checkpoint to the reviewer.

Round 3

Reviewer 1 Report

Thanks for the clarification.